# A Comprehensive Review of the Effects of Hyoscine Butylbromide in Childhood

**DOI:** 10.3390/jcm14093009

**Published:** 2025-04-26

**Authors:** Rodrigo Vázquez Frias, André Hoerning, Christian Boggio Marzet, Martin C. Michel

**Affiliations:** 1Research Vice Director Office, Hospital Infantil de México Federico Gómez, Mexico City 06720, Mexico; rovaf@yahoo.com; 2Department of Pediatrics, Pediatric Gastroenterology, Hepatology and Endoscopy, Erlangen University, 91054 Erlangen, Germany; andre.hoerning@uk-erlangen.de; 3Pediatric Gastroenterology and Nutrition Working Group, Hospital Gral. de Agudos “Dr. Ignacio Provano”, Buenos Aires 1426, Argentina; cboggio35@hotmail.com; 4Department of Pharmacology, University Medical Center, Johannes Gutenberg University, 55131 Mainz, Germany

**Keywords:** hyoscine butylbromide, children, infants, abdominal cramps, abdominal pain, vomiting, childbirth

## Abstract

**Background/Objectives**: Hyoscine butylbromide (HBB) is a spasmolytic drug classified as indispensable by the World Health Organization. While mostly used in adults, it is also approved for use in adolescents and children aged 6 years and older. We have comprehensively reviewed the efficacy and safety of HBB in approved and off-label childhood indications. **Results**: Childhood studies covered an age range starting as early as 2 days. A randomized controlled trial (RCT) found a similar efficacy compared to paracetamol in the approved indication of abdominal cramps and pain. Among off-label uses, several studies demonstrate efficacy in general anesthesia and various diagnostic procedures, but the largest body of evidence relates to use in childbirth/labor, including 17 RCTs. While these largely focused on efficacy outcomes on the mother, fetal safety outcomes were reported in 12 of these studies, mostly as effects on the APGAR score and/or heart rate. The overall evidence supports safety in infants and children including those younger than the approved use age of 6 years and older. **Conclusions**: While only limited pediatric efficacy data from RCTs are available in the approved indications, data from thousands of patients in RCTs, case series, and non-randomized trials do not raise concerns on the safety and tolerability of HBB in childhood. Additional dedicated childhood studies, particularly RCTs, on efficacy are recommended.

## 1. Introduction

Hyoscine butyl bromide (HBB, also known as butylscopolamine bromide or scopolamine butylbromide) is a spasmolytic drug due to being a muscarinic receptor antagonist with some additional ganglionic blocking effects [1]. Based on its quaternary amine structure, it is unlikely to pass the blood–brain barrier, with an anticipated benefit of lacking cognitive side effects that are typical for muscarinic antagonists with a tertiary amine structure [2]. The primary approved indication of HBB is the treatment of abdominal cramps and pain, for which it is recommended in various guidelines as reviewed elsewhere [3,4]. Other indications, for which it is approved in some countries, include irritable bowel syndrome, genito-urinary spasms, and dysmenorrhea. Accordingly, the World Health Organization lists HBB as an indispensable drug. HBB is approved for use in adults and in adolescents and children aged 6 years and older in most countries; some fixed dose combinations [5,6], e.g., HBB in combination with paracetamol (also known as acetaminophen), are approved for use in patients aged 12 years and older.

Against this background, we comprehensively reviewed data on the efficacy and safety of HBB in approved and off-label uses in infants, children, and adolescents across various indications. We considered randomized controlled trials (RCTs), observational studies and, particularly from the early literature, case series.

## 2. Materials and Methods

Our analysis is based on a PubMed search using the key word combination of “children” with “hyoscine butyl bromide” or “butylscopolamine bromide” or “scopolamine butylbromide”. Additional references were retrieved from the databases of Opella, a Sanofi company (Frankfurt, Germany), the marketing authorization holder of HBB in various countries. All retrieved references were analyzed and included in the manuscript if they reported on the use of HBB in infants, children, and adolescents. In some cases, we could retrieve the abstract but not the full paper; these typically were published in journals that do not exist electronically, at least not for those older publication years. As these were typically reporting on studies not sponsored by Sanofi–Aventis, Sanofi–Aventis also had no full text versions of these available. Requests to obtain them via the institutional library also failed. Based on the poor reporting quality of the older studies (none of them an RCT; see Section 3), we concluded that these were unlikely to change the overall conclusions of the manuscript. Therefore, the information in the retrieved abstracts was interpreted very conservatively unless quantitative data were presented and no major conclusions were based on these studies. Following a summary of early reports, we will divide our analysis into indications involving prolonged treatment (days to weeks) and acute treatments (single day).

## 3. Early Reports in Children

Reports on the use of HBB in children go back for almost 70 years. As typical for articles from that time, the technical and reporting quality is limited. For instance, they often reported on a mix of several indications and talked generally about infants and children without specifying the age of the patients being treated. Moreover, the definition or naming of conditions at the time may have deviated from currently applied definitions and terminology. Finally, most of them are single-center case series without a control group and therefore carry a substantial risk of bias. Some early reports were identified, mostly from the reference lists of other articles, but neither print copies nor electronic publications and not even abstracts could be retrieved [7,8,9,10].

Japanese investigators reported on the beneficial effects of HBB tablets or suppositories in a total of 17 children including six (aged 33–148 days) with vomiting due to pyloric spams, four (aged 3.3–14.6 months) with gastric pain due to Moro’s intermittent umbilical cord colic, three (aged 2.1–3.4 years) with coughing, one baby with diarrhea, and three children (aged 5.8–9.6 years) with nephritis [11]; the efficacy was described as good to excellent in most cases, and some of them were supported by X-ray investigation; no major effects were observed in conjunctival blood vessels. No information on tolerability was provided.

Austrian investigators studied a total of 88 infants and children with vomiting, acute dyspepsia, and children that had recovered from dyspepsia but had remaining symptoms of intestinal hypermotility, with “constitutional intestinal motility disorders”, or with spastic constipation [12]. Efficacy was observed in most patients but described only qualitatively as “good results”; this manifested not only as symptom improvement but, in the case of dyspepsia, as a shorter time to resolution of the symptoms. The authors stated that HBB was “devoid of side effects”, particularly those observed with atropine, but did not provide specific information.

Another group of Japanese investigators reported on 38 cases (eight with pyloric spams, five with infantile colic, 15 with enteritis, two with acute dyspepsia, two with dysentery, six others) which received parenteral administration, suppository, or oral treatment in 24, nine, and two cases, respectively [13]. They reported efficacy in a semi-quantitative manner as full, partial, and lack of relief of symptoms. This included eight cases with pyloric spasms (aged 1–3 months old), most of whom had failed other treatment efforts including scopolamine or atropine and in two cases symptoms were so severe that their body weight at the age of 1 month was less than at birth; HBB was effective in all infants, and no relevant side effects were observed. Five patients with severe infantile colic (aged 5 years and 8 months to 12 years and 10 months) had failed to respond to scopolamine, but HBB was effective in all of them. Among 15 cases of acute enteritis (aged 4–14 months), 11 exhibited full and four partial relief; resolution of the symptoms compared within 1–2 days as compared to 3–4 days in historic controls. Two patients each with dyspepsia (aged 2–14 months) and dysentery (aged 20–26 months) also responded well to HBB. Finally, HBB was also effective in six patients with other gastrointestinal conditions (aged 4 days to 13 years). In the overall group of 38 children, full, partial, and lack of symptom relief was observed in six, 25, and seven cases, respectively. Of note, side effects were not observed in any of these infants and children.

Investigators from Argentina reported on the use of HBB in hypertensive vesicular dyskinesia of the gallbladder in five children (aged 7–9 years) combining clinical and radiological assessment (oral cholecystography) [14]. The symptoms disappeared in all cases after the administration of HBB and a repetition of the radiological evaluation confirmed a reduced contraction intensity of the gallbladder. Adverse events were not observed.

One group from Mexico reported on the treatment of 124 children (aged 2 days to 6 years) with oral HBB solution (3–5 mg/kg), including 51 with colic and hiccups, 35 with vomiting, 21 with transient dyspepsia, and 17 with constipation [15]. While efficacy data were not reported quantitatively, they were rated as very good by the investigators in 96.8% of cases. No information on adverse effects was provided.

Other Mexican investigators reported on a series of 240 children (aged 1 month to 12 years) including 18 with transient dyspepsia, 39 with gastritis, 24 with gastroenteritis, 67 with ileo-colitis, and 71 with colitis [16]. The treatment efficacy in transient dyspepsia was reported only qualitatively as very good in 100% of cases; in gastritis as very good, good, or ineffective in 86.4%, 11.9%, and 1.7% of cases, respectively; in gastroenteritis as very good, good, or ineffective in 41.6%, 52.2%, and 4.2% of cases, respectively; in ileo-colitis as very good, good, or ineffective in 89.6%, 9.0%, and 1.4% of cases, respectively; and in colitis as very good, good, or ineffective in 94.4%, 2.8%, and 2.8% of cases, respectively. Twenty patients with intestinal parasitosis received HBB prophylactically and reported no gastrointestinal complaints during antiparasitic treatment. None of the 240 children reported side effects. The authors concluded that at least satisfactory results were observed in 97.9% of patients with no undesirable side effects.

Thus, early reports on HBB in infants and children came from multiple Asian, European, and Latin American countries focused mostly on use in functional gastrointestinal disorders. Generally, they reported promising efficacy but provided little quantitative information and, in several cases, lacked information on safety and tolerability, thus, overall, not generating robust evidence. However, they indicate that HBB has successfully been used in infants (starting at about 30 days of age) and children for seven decades apparently without overt side effects, mostly in gastrointestinal indications.

## 4. Studies on Prolonged Use in Childhood

This section covers indications where HBB is administered for at least several days.

### 4.1. Gastrointestional Cramps and Pain

Gastrointestinal cramps and pain are the primary indication for use of HBB in adults, for which it is recommended in guidelines from various countries [3,4]. Following the mixed indication studies from the 1950s and 1960s (see above), several dedicated studies were performed in this indication. A superiority trial randomized 236 children (aged 8–17 years) with non-specific colicky abdominal pain presenting in a pediatric emergency department in Canada to receive either HBB or paracetamol [17]. The primary endpoint was self-reported pain intensity on a 100 mm visual analog scale as assessed 80 min after drug administration. The minimally clinically important difference on this scale was 13 mm. The observed improvements were much larger with both medications, i.e., 29 ± 26 and 30 ± 29 mm in the HBB and paracetamol group, respectively (adjusted difference 1 mm [95% confidence interval −7 to 7]; Figure 1). Rescue analgesia was administered to four participants in the HBB and one in the paracetamol group. The proportion of patients with a pain score less than 30 mm was 55.0% and 54.3%. The incidence of adverse events was also similar in both groups (27.6% vs. 24.3%), and no serious adverse events were reported. The authors concluded that both medications caused clinically important pain reduction, but superiority relative to paracetamol was not established in non-specific colicky abdominal pain.

A multi-center observational study in Germany compared a multi-component homeopathic preparation (“Spascupreel”) to HBB administered for 1 week in 204 children aged 12 years or younger with newly diagnosed or recurring gastrointestinal or urethral spasms. Treatment allocation was at the discretion of the investigator. Outcome measures were the severity of the spasms and clinical symptoms (pain/cramps, sleep disturbance, distress, eating or drinking difficulties, and frequent crying) as well as patient-reported compliance global tolerability [18]. In this observational study, the two treatment groups differed in age (7.6 ± 2.7 vs. 9.2 ± 2.2 years of age with Spascupreel and HBB, respectively), height, and body weight; moreover, the Spascupreel group more frequently included children with urinary tract complaints (20 vs. 5%) and exhibited a lesser degree of sleep disturbances and crying, but a greater fraction of patients with an assumed psychological basis of symptoms (9 vs. 3%). With these caveats, both treatments exhibited a comparable efficacy across all efficacy and tolerability parameters.

### 4.2. Vomiting

Some early case series reported on patients with vomiting as part of case series including multiple conditions, age groups, and HBB formulations [11,12,15] (see above).

An Italian group reported on 15 cases of vomiting due to various causes (age neonatal with birth weight of 1700 g to 15.5 months) treated either with intramuscular injections (n = 3) or suppositories of HBB (n = 11); one child started on injections and was converted to suppositories [19]. Among that, eight cases with pyloric spasms all experienced a “positive outcome” with a “significant reduction in number of vomiting episodes” (no quantitative information provided) with improvement typically occurring within 3–6 days of treatment. Among the three cases with pyloric spasms and radiological evidence of moderate hypertrophy, vomiting subsided within 2–8 days of treatment and disappeared completely within 30 days. In the four cases with vomiting during or after gastrointestinal disorders, a reduction in vomiting episodes was observed on the first or second day of treatment and disappeared completely within 3–4 days. A resumption of weight gain was noted in all 15 infants. No information on tolerability was provided.

While these early results look promising, we have not identified additional reports on the effects of HBB on vomiting in infants and children, particularly any published less than 50 years ago. Importantly, we have not identified any controlled study reporting vomiting as an outcome parameter.

### 4.3. Enuresis

Nocturnal enuresis is a common condition of childhood and is generally attributed to disturbances in the circadian rhythm of arginine vasopressin leading to nocturnal polyuria; therefore, treatment with vasopressin or its analogs is the current medical approach of choice [20]. An early report from Spain described a series of 30 children with enuresis with short descriptions of each child but no summary analysis of the data. The authors concluded that HBB may be promising but requires additional investigation [21].

Israeli investigators performed a double-blind cross-over study in 20 children with enuresis comparing HBB with imipramine and with a placebo [22]. With six children dropping out (apparently unrelated to adverse events), 14 were available for analysis. The treatment duration was 28 days with each drug in the first 7 and 21 days in the subsequent patients; the latter received a doubling of the imipramine dose in the last week of observation. Data were analyzed as the number of nights with enuresis relative to nights observed. The total number of enuresis episodes was 188 with the placebo, 180 with HBB, and 128 in the imipramine treatment period. Tolerability data were not reported. Based on these disappointing results for HBB, apparently no additional studies were conducted in this indication.

## 5. Studies on Acute (Single Dose) Use in Childhood

While acute (single day, mostly single dose) uses of HBB have been studied frequently, all indications covered in these trials are off-label uses.

### 5.1. General Anesthesia

Vagal reflexes can lead to bradycardia during general anesthesia, and muscarinic receptor antagonists including atropine and hyoscine have been used in adults to prevent this during general anesthesia. Moreover, bradycardia can also occur when cholinesterase inhibitors are used to terminate the effects of skeletal muscle relaxants such as succinyl choline. British investigators randomized 100 consecutive patients undergoing general anesthesia to receive i.v. injections of either atropine or HBB [23]. The main outcome parameter was audible respiratory secretions that were scored semi-quantitatively as none, audible but no intervention required, audible and one suction required to retain smooth aspiration, or repeated suctions required, as scored by an investigator blinded to treatment allocation. While HBB was superior to atropine in the 40 and 38 adult patients, respectively, it was similarly effective in the 11 children each. Based on an analysis of heart rate and ECG patterns, HBB provided better protection against vagal slowing of the heart (no separate analysis of adult and pediatric patients was provided).

A Japanese group has published four consecutive reports related to the use of HBB in general anesthesia. The first report assessed the effectiveness of HBB in preventing edrophonium-induced bradycardia in 50 infants of children occurring when reversing the effects of tubocurarine [24]. In total, 25 children each received either edrophonium plus atropine or plus HBB. The difference between the highest and lowest heart rate within a child was 54 ± 21 bpm with atropine and 34 ± bpm with HBB. Mean heart rate as observed over 5 min after administration of atropine dropped considerably whereas it did not upon the use of HBB. Thus, the authors concluded that HBB was more effective in preventing initial bradycardia caused by edrophonium. A follow-up study included boys and girls (15 each) with a mean age of 5.3 ± 2.2 years undergoing halothane anesthesia [25]. While heart rate and mean blood pressure decreased following the induction of halothane anesthesia, heart rate increased above the pre-induction values after HBB and remained elevated throughout the observed period. Blood pressure reached pre-induction levels within 2 min after administration of HBB. The authors highlighted the rapid onset of action of HBB as a potential benefit. A third report focused on echocardiography assessed cardiac function during halothane anesthesia [26]. HBB was found to attenuate the halothane-associated deterioration of the pre-ejection period and systolic time interval but did not improve rate-independent indices of cardiac function. Based on such data, a fourth study was conducted that randomized 42 infants and children undergoing halothane anesthesia to receive 0.2 mg/kg or 0.4 mg/kg HBB in conjunction with succinylcholine (2 mg/kg) [27]. After heart rate had declined with halothane anesthesia, both doses of the HBB/succinyl choline combination increased heart rate above pre-induction values within 20 s of injection; a decrease in heart rate was not observed at any time point. The two doses of HBB were equally effective in this regard. Although the interpretation of these data is limited by coming from a single center, the data indicate that HBB is effective in preventing bradycardia and may have advantages over atropine in infants and children. Neither report included tolerability data, and no later reports were identified.

### 5.2. Diagnostic Use

Four articles reported on the use of HBB in diagnostic procedures. One German study described the use of i.v. administration of HBB in 22 children undergoing radiological examination of their esophagus [28]. They found that varices of the esophagus were shown “much better than with conventional techniques” and that small axial hiatus herniae and esophageal stenoses were better shown. Others reported on the use of HBB in double contrast examination of the stomach [29]. Japanese investigators reported the use of HBB and prostaglandin F_2α_ in anorectal manometry in children with Hirschsprung disease [30]. While the prostaglandin facilitated normal rectoanal reflexes in six out of seven children, it was abolished or markedly inhibited in 12 out of 13 children by HBB. They concluded that both agents facilitated the manometric diagnosis of the condition. A Canadian radiologist described the protocol to perform magnetic resonance enterography to diagnose Crohn’s disease in children in her institution [31]. She reported that both glucagon and HBB can be helpful for reducing peristalsis. In a series of 50 consecutive children, about 20% experienced mild blurred vision, tachycardia, or a dry mouth, and <10% reported vomiting after receiving HBB. She concludes that this tolerability profile is better than that of glucagon, making HBB the agent of choice for this procedure at her institution. Finally, a case report on a 14-month-old boy described how HBB facilitated endoscopic retrograde cholangiopancreatography in this child [32].

### 5.3. Childbirth/Labor

Many reports relate to the use of HBB during childbirth/labor. In this case, the mother and not the unborn is the efficacy target of the drug. Thus, apparently more studies were reported for the childbirth/labor indication than that of all other uses combined despite this being an off-label indication. Early studies in the field were largely based on single center cohorts and often did not report quantitative data and/or focused on maternal outcomes [33,34,35,36,37,38,39,40,41,42,43,44,45,46,47,48,49,50,51,52,53,54,55]; these will not be discussed further.

More relevant information comes from 17 RCTs and two non-randomized controlled trials. We will focus here on the studies that reported on fetal outcomes and ignore those that focused on maternal outcomes [56,57,58]. These maternal outcomes have been described and discussed in various systematic reviews and meta-analyses [59,60,61,62], including one from the Cochrane collaboration [63]. The maternal efficacy of HBB during childbirth and labor is supported by various experimental studies [64,65,66,67,68]; these will not be discussed here further.

The most frequently reported fetal outcome parameter upon administering HBB during childbirth/labor is the APGAR score (12 studies); therefore, data on this parameter are summarized in Table 1. The earliest RCT in the childbirth/labor indication randomized 129 women from Jamaica to receive i.v. HBB or placebo [69]. The median APGAR scores noted at 1 and 5 min after delivery did not differ (9 in both groups at both time points). An RCT from India randomized 146 women to received i.v. HBB, i.m. drotaverine, or a placebo [70]. The APGAR scores were 8 and 9 at 1 and 5 min, respectively, in all three groups. Admissions to the neonatal intensive care unit occurred in one, two, and three newborns, most likely not reflecting drug effects. Another Indian study randomized 104 primigravidae to i.v. HBB or saline [71]. While one neonate in the HBB group had an APGAR < 7 at 1 min, it fared well subsequently; all babies had APGAR scores > 7 at 5 min with a median of 9 in both groups. No baby in either group required admission to the neonatal unit. An Iranian study randomized 130 women to receive an HBB suppository or placebo [72]. Fetal heart rate (141 vs. 138 bpm) and heart rate variability were similar in both groups. The APGAR score was 8–10 in both groups at 1 and 5 min, but two children in the HBB group had a score of 4–6 at 1 min. Investigators from Saudi Arabia randomized 97 women who received i.m. HBB or a placebo [73]. A total of 8 of 37 babies in the placebo and 5 of 47 in the HBB group had an APGAR score of <9 whereas all others had scores of 9–10. Another Iranian study randomized 188 multipara to receive i.v. HBB or saline [74]. The APGAR score at 1 min was 8.4 ± 1.6 and 8.1 ± 1.8, respectively, and at 5 min 7.8 ± 1.2 and 8.1 ± 1.9, respectively. Two neonates in the HBB and one in the saline groups required resuscitation, and one in each group admission to the neonatal intensive care unit; none of these group differences reached statistical significance. Turkish obstetricians randomized primigravid and multigravida women to receive i.v. HBB or a placebo [75]. The APGAR scores were 8.09 ± 0.41 and 8.16 ± 0.3 at 1 min and 9.11 ± 0.38 and 9.16 ± 0.35 at 5 min. Nigerian investigators randomized 160 women to receive i.v. HBB or saline [76]. APGAR scores were 8.08 ± 1.54 and 7.64 ± 1.60 at 1 min and 9.54 ± 1.09 and 9.40 ± 1.09 at 5 min. Fetal tachycardia was observed in one and two cases in the HBB and saline groups, respectively. Other obstetricians from Nigeria randomized 126 nulliparous women to i.v. HBB or a placebo [77]. APGAR scores < 7 were observed in 15.9% and 20.6% after 1 min in the HBB and placebo groups, respectively, and in 1.6% and 6.3% at 5 min. Admission to the neonatal intensive care unit occurred in five and four cases. The fetal heart rate was 141.4 ± 0.1 in the HBB and 141.0 ± 10.7 in the placebo groups. Norwegian physicians randomized 249 nulliparous women to receive i.v. HBB or saline [78]. An APGAR score < 7 was observed in 0.8% in the HBB group but not in the placebo group after 5 min. Median fetal heart rate did not change in either group. Admission to the neonatal intensive care unit occurred in 2.4% and 3.2%.

Two RCTs were designed differently: A group from Nigeria randomized 126 women to receive i.m. HBB or a placebo with both groups additionally receiving 25 µg vaginal misoprostol [79]. The APGAR score at 1 min was 7.35 ± 1.43 and 7.03 ± 1.54 with HBB and the placebo, respectively, and at 5 min 8.54 ± 1.54 and 8.54 ± 1.20. An APGAR score < 7 occurred in 12.7% and 22.2% at 1 min and in 6.3% and 7.9% at 5 min in the HBB and placebo groups, respectively. Admission to the neonatal unit occurred in 9.5% and 20.6%. While most of the above studies had used a single dose of 20 mg HBB, an RCT from Egypt assigned 120 primigravida to receive 20 mg HBB, 40 mg HBB, or a placebo i.v. [80]. Mean APGAR scores at 1 min were 7.1 to 7.2 and at 5 min 8.1 to 8.2.

Finally, a study from Iran randomized 120 women to receive i.v. HBB or atropine, i.e., the study did not include a placebo arm [81]. The APGAR score at 1 min was 8.9 ± 0.4 with HBB compared to 8.88 ± 0.4 with atropine; the APGAR score at 5 min was also similar (no quantitative data shown). Taken together these studies demonstrate consistently that a single dose of HBB administered during labor does not affect the APGAR score as an indicator of neonatal wellbeing. Although studied in fewer cases, HBB also did not affect neonatal heart rate. Accordingly, admission rates to the neonatal care units were also not affected.

The oldest non-randomized but controlled study based on a series of 100 women from the UK reported that HBB affected neither rate nor rhythm of the fetal heartbeat but did not provide quantitative data [40]. In a non-randomized series of 200 women from India, three babies with HBB and five in the control groups had APGAR scores of 4–6 at 1 min [82]. They were kept in a nursery under observation for 24 h and were then discharged in good condition. The APGAR score at 5 min was 8–10 in all 200 babies. Overall, HBB has been studied in both nulli- and multiparous women (Table 1). While only one report described data on ethnicity, the fact that the studies came from many countries (three studies from Nigeria, two studies each from India and Iran, one study each from Jamaica, Norway, Saudi Arabia, Türkiye, and the UK), we conclude that HBB is equally safe across ethnicities in the labor/childbirth indication.

### 5.4. Other Acute Uses

Several reports on the pediatric use of HBB were retrieved that dealt with a condition not explored in any other study but mostly dealing with the gastrointestinal tract. Some of these represent case reports.

Greek investigators reported on the use of HBB to assist in the removal of swallowed objects stuck in the esophagus of 79 children (aged 8 months to 13 years) [83]. Based on the initial experience, a prospective study in 11 children (aged 10 months to 10 years) was designed, in which the foreign body passed to the stomach within the first six hours; it passed through the gastrointestinal tract and was excreted within 1–3 days. However, these attempts were unsuccessful in two children, yielding an overall success rate of 82%. In a related approach, British physicians reported a retrospective chart study of 46 patients (aged 7–96 years including 5 younger than 20 years) with food bolus impaction of the esophagus; however, they found a similar relief rate with and without the use of HBB [84].

Investigators from Pakistan conducted a non-randomized comparison of providing i.v. fluids, i.v. HBB, and hypertonic saline enema in a total of 45 children with a diagnosis of intestinal obstruction due to *Ascaris lumbricoides* [85]. While HBB provided faster relief than fluid administration, the fastest relief was observed with the hypertonic saline enemas. Turkish pediatric surgeons communicated a case report in which a child with biliary ascariasis clinically manifesting with pain, vomiting, and abdominal tenderness was successfully treated with a combination of mebendazole and HBB [86].

## 6. Discussion and Conclusions

HBB is approved for the treatment of gastrointestinal and genito-urinary cramps and pain not only in adults but also in children and adolescents aged 6 years and older in most countries; some additional indications are available in a more limited number of countries. To obtain a comprehensive picture of the use of HBB in infants and children, we have retrieved many case series and randomized and non-randomized controlled studies. These included approved indications and off-label uses (Figure 2).

The first reports on the use of HBB in children appeared 70 years ago, indicating a long history of clinical experience with this drug in children and adolescents, and to a more limited extent even in infants. However, most of the early publications were based on case series, very often in a mixed group of conditions and without quantitative data. In line with the approved indication of HBB, these early reports focused on gastrointestinal indications. In the absence of quantitative data, limited conclusions can be drawn from these reports except that HBB has been in pediatric use for a long time and was generally considered safe.

As often seen in pediatrics, there is a limited number of RCTs on the use of HBB in infants, children, and adolescents. For instance, we found only a single RCT testing HBB in prolonged use in a gastrointestinal indication, i.e., the primary approved indication, and one in children with enuresis (not an approved indication).

More RCTs were found for uses involving only short-term, often single-dose administration of HBB, e.g., in anesthesia. However, we identified 17 RCTs and two non-randomized control trials for the use of HBB during childbirth/labor. While most of these primarily focused on the mother, 12 included data on the safety of the neonate as assessed by APGAR scores and, in some cases, heart rate. The latter studies indicate that HBB administered a few times on a single day has no overt adverse effects on the neonate as the probably most vulnerable group of patients.

Thus, the available data on the use of HBB in infants, children, and adolescents provide only limited robust evidence on its efficacy in various indications. However, the efficacy of HBB at least in the gastrointestinal indications has been documented in various studies in adults, where HBB is recommended in various guidelines [3,4]. Accordingly, HBB is considered to be an indispensable drug by the World Health Organization. Thus, prescription data from Bahrain indicate that HBB is one of the most frequently used drugs by general practitioners in that country [87]. The limited availability of RCTs in children indicates that the regulatory approval of HBB in children and adolescents was largely based on the studies in adults and tolerability data in children and adolescents, as well as its long-standing clinical use.

The perhaps most interesting finding of our analyses is that most published studies relate to off-label uses of HBB. This can be an off-label indication including use in general anesthesia, diagnostic procedures, and most importantly during childbirth. Despite being an off-label use, the administration of HBB during childbirth has apparently become so standard that some trials used it as background treatment, for instances when comparing the presence or absence of a family member during labor [88] or exploring the analgesic effect of exercise during childbirth [89].

The other aspect of off-label use with children is that its regulatory approval is for children and adolescents aged six years and older. However, many of the studies reviewed here included or even focused on much younger children, with the youngest exposed infant (other than in the childbirth studies) being only two days old. While particularly the short-term studies are insufficient to robustly demonstrate safety and tolerability in these very young children, it is noteworthy that no relevant safety concerns have emerged. However, a case report describes a 4-year-old girl who accidentally ingested 12 tablets of 150 µg each and presented to an emergency department with hyperactivity and confusion; the patient improved and could be discharged after 2 days at the hospital without lasting consequences [90]. Other than this overdose, no cases of severe adverse events were found in our searches.

Reports on the use of HBB in infants, children, and adolescents come from a wide range of countries in Asia (India, Japan, Pakistan), Africa (Nigeria), Europe (Austria, Denmark, Germany, Greece, Italy, Norway, Spain, Türkiye, United Kingdom), Latin America (Argentina, Jamaica, Mexico) and North America (Canada), and the Middle East (Egypt, Iran, Israel, Saudi Arabia). This wide geographical distribution provides indirect evidence that the efficacy and tolerability of HBB applies to various ethnicities and can be observed in various healthcare settings.

In conclusion, our data indicate that HBB is successfully applied in various conditions with many of these indications and age groups being outside the approved uses as defined in the prescribing information. While the aggregate evidence indicates that HBB is safe across these indications and age groups, properly designed studies are recommended and needed to substantiate these conclusions. Unfortunately, based on clinicaltrials.gov (assessed on 5 April 2025), only one study of HBB in children is currently ongoing; this trial compares HBB with onabotulinum toxin A for the treatment of drooling in children with cerebral palsy (NCT03616067).

## Figures and Tables

**Figure 1 jcm-14-03009-f001:**
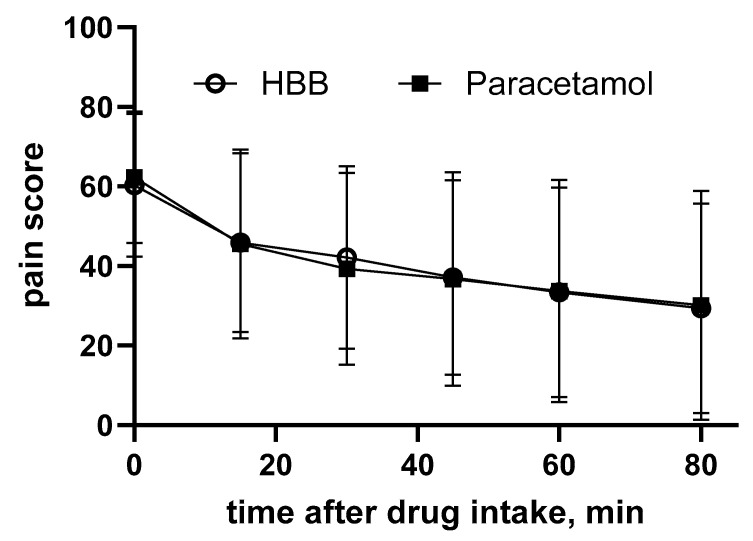
Comparison of effects of HBB and paracetamol on non-specific colicky abdominal pain in a head-to-head RCT. Data are shown as means ± SD of the pain score (primary study endpoint) measured on a visual analog scale ranging from 0 to 100. Adapted from [17].

**Figure 2 jcm-14-03009-f002:**
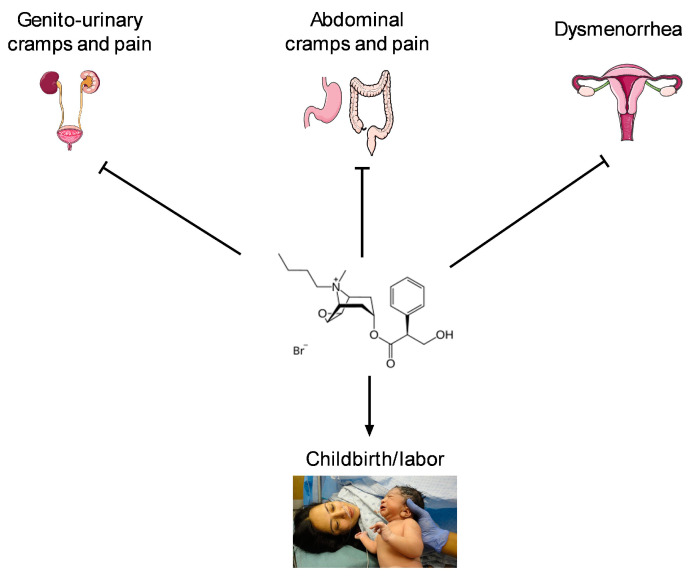
Approved (**upper row**) and off-label but intensively studied (**lower row**) indications where HBB inhibits (**upper row**) or promotes (**lower row**) an effect. For details see main text. Images used are from Servier Medical Art (SMART) (**upper row**) and courses.lumenlearning.com (**lower row**) under CC-BY 4.0.

**Table 1 jcm-14-03009-t001:** Effect of HBB administered during childbirth on neonatal APGAR scores in placebo/saline controlled RCTs as assessed after 1 min. Data are mean or median as reported by the investigators; n.r.: not reported quantitatively for % nulliparous in the control and HBB groups or not reported as specific value but statement on not being different between groups in the text of the article. A single dose was administered unless stated otherwise. Mean/median age of the mother ranged from 22 to 30 years.

Study	% Nulliparous	HBB Dose	Route	APGAR Score	
				Placebo	HBB
Samuels 2007 [69]	58/48	20 mg	i.v.	9	9
Gupta 2008 [70]	55/45	20 mg per 30 min (max 3 doses)	i.v.	8	8
Aggarwal 2008 [71]	n.r./n.r.	40 mg (mostly every 2–4 h)	i.v.	n.r.	n.r.
Makvandi 2011 [72]	100/100	20 mg	rectal	n.r.	n.r.
Qahtani 2011 [73]	n.r./n.r.	40 mg	i.m.	n.r.	n.r.
Sekhavat 2012 [74]	0/0	20 mg	i.v.	8.1	8.4
Kirim 2015 [75]	46/48	20 mg	i.v.	8.16	8.09
Imarulu 2017 [76]	<50/<50	20 mg	i.v.	7.64	8.08
Akiseku 2021 [77]	100/100	20 mg	i.v.	n.r.	n.r.
Gaudernack 2024 [78]	100/100	20 mg	i.v.	n.r.	n.r.
Agadaga 2024 [79]	57/62	20 mg	i.m.	7.03	7.35
Maged 2018 [80]	100/100	20 or 40 mg	i.v.	8.1	8.2 *

* 8.1 in the 40 mg group.

## Data Availability

No new data were created or analyzed in this study. Data sharing is not applicable to this article. Articles used in this work, regardless of their source, may be provided or shared by the authors upon reasonable request as far as copyright law allows.

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
