# Peer review of "A Comprehensive Review of the Effects of Hyoscine Butylbromide in Childhood"

_jcm, 2025, doi:10.3390/jcm14093009_

Round 1

Reviewer 1 Report

Comments and Suggestions for Authors

The manuscript by Vázquez Frias et al. addressed the safety and tolerability of HBB in childhood. The authors describe several studies and the results of each related to the use of HBB in children. Below are some comments the authors should address:

The headings should be removed from the abstract.

Were there specific inclusion and exclusion criteria for the selection of articles used in this work? If so, please detail them in the appropriate section.

According to the SANRA scale (Res Integr Peer Rev 4, 5; 2019), a compelling narrative review should be transparent about the sources of information on which the text is based. Given that many of the articles are not easily accessible online, could the authors share additional references from the Sanofi-Aventis databases (Line 55) as supplementary material?

Lines 208-211: What is the reference for this statement?

Lines 263-266: How did HBB facilitate endoscopic retrograde cholangiopancreatography?

Given that the authors mention that the mother, not the fetus, is the target of HBB efficacy (Lines 268-269), is it plausible to consider the characteristics of the mothers in the studies in this section? For example: race, age, number of previous births, medical history, etc. This information could be added to Table 1.

The literature reports that cases of HBB toxicity are primarily unintentional (BMJ Case Rep. 2020;13(2):e234029; Journal of the Royal Society of Health. 1997;117(4):242-244). However, within the authors' literature search, were there any studies showing fatal outcomes due to standard HBB use? If so, this should be clarified in the manuscript.

Finally, this reviewer would like the authors to expand some of the scope or provide a clarifying summary table that could guide additional direct questions requiring prioritization of future research and identify key knowledge gaps that could be addressed.

Author Response

General comment: The manuscript by Vázquez Frias et al. addressed the safety and tolerability of HBB in childhood. The authors describe several studies and the results of each related to the use of HBB in children. Below are some comments the authors should address:

Reply: We appreciate the time you invested in reviewing our manuscript and in providing relevant comments.

Comment 1: The headings should be removed from the abstract.

Reply 1: Journals have different styles and preferences how an abstract should be written. Your opinion on this is appreciated. However, these subheadings within the abstract are based on the journal style (https://www.mdpi.com/journal/jcm/instructions#front). Therefore, they were not removed to comply with the journal style.

Comment 2: Were there specific inclusion and exclusion criteria for the selection of articles used in this work? If so, please detail them in the appropriate section.

Reply 2: We did not exclude any retrieved publication as long as it reported on the use of HBB in infants, children and adolescents. The clarify this, the following text was added to section 2: “All retrieved references were analyzed and included in the manuscript if they reported on the use of HBB in infants, children and adolescents.”

Comment 3: According to the SANRA scale (Res Integr Peer Rev 4, 5; 2019), a compelling narrative review should be transparent about the sources of information on which the text is based. Given that many of the articles are not easily accessible online, could the authors share additional references from the Sanofi-Aventis databases (Line 55) as supplementary material?

Reply 3: We appreciate the reviewer’s emphasis on transparent information sharing and fully agree with this intention. Only transparent information sharing builds trust between authors and readers. We understand from the reviewer comments, that he/she is not concerned about lack of transparency which sources of information were used (all are fully referenced). We agree that getting full texts of various references turned out to be difficult or even impossible, particularly for some of the older studies that neither exist in electronic databases nor are available electronically in online versions of the respective journals. As already stated in the manuscript (last sentence of first paragraph of section 3), there are four early publications (references 7-10) for which we could not retrieve a full text and not even an abstract - even with the support of Sanofi-Aventis. There are some other publications for which we could retrieve the abstract but not the full paper; these typically were published in journals that do not exist electronically, at least not for those older publication years. As these were typically reporting on studies not sponsored by Sanofi-Aventis, Sanofi-Aventis also had no full text version of these available. Requests to obtain them via the institutional library also failed. Based on the poor reporting quality of the older studies (none of them an RCT; see section 3 of our manuscript), we concluded that these were unlikely to change the overall conclusions of the manuscript. Therefore, the information in the retrieved abstracts was interpreted very conservatively unless quantitative data were presented, and no major conclusions were based on these studies. To make this more transparent, the following text was added to section 2: “In some cases, we could retrieve the abstract but not the full paper; these typically were published in journals that do not exist electronically, at least not for those older publication years. As these were typically reporting on studies not sponsored by Sanofi-Aventis, Sanofi-Aventis also had no full text version of these available. Requests to obtain them via the institutional library also failed. Based on the poor reporting quality of the older studies (none of them an RCT; see section 3), we concluded that these were unlikely to change the overall conclusions of the manuscript. Therefore, the information in the retrieved abstracts was interpreted very conservatively unless quantitative data were presented and no major conclusions were based on these studies.”

The referee further suggests to make the references “not easily accessible online” available as online supplement. Unfortunately, we cannot accommodate that requests for multiple reasons: Firstly, and most importantly, published articles are copyrighted and must not be reproduced without explicit consent of the copyright holder; doing otherwise would expose us and the publisher to potential legal consequences. Second, to the best of our knowledge, adding full papers being referenced as online supplement has never been done before in any journal we are aware of. Third, “not easily accessible online” is a vague term. In our experience, whether and how difficult we could obtain full texts did not necessarily correlate whether a reference was obtained from Sanofi or came up in our own searches. While this response may be frustrating, we will happily make full texts that we have available to the reviewer offline if he/she provides us with a list he/she specifically would be interested in.

Comment 4: Lines 208-211: What is the reference for this statement?

Reply 4: This statement is found in various forms in most articles cited in section 5.1  and not referenced in any of them (all published in anesthesiology journals), implying that it may be textbook knowledge among anesthesiologists. Therefore, we have added a general review on the topic as additional reference (Doyle 1990).

Comment 5: Lines 263-266: How did HBB facilitate endoscopic retrograde cholangiopancreatography?

Reply 5: We apologize for the confusion, but this was a typo. “how” should have read “that”, which was corrected. The case reported cited here did not provide any underlying mechanistic information and only speculated on suitable HBB dose in an infant.

Comment 6: Given that the authors mention that the mother, not the fetus, is the target of HBB efficacy (Lines 268-269), is it plausible to consider the characteristics of the mothers in the studies in this section? For example: race, age, number of previous births, medical history, etc. This information could be added to Table 1.

Reply 6: You raise an interesting question, and we have extracted the corresponding information from all studies in Table 1. Only one of the studies specifically mentioned ethnicity (Samuels 2007 saying “mostly Afro-Caribbean” without providing specific numbers). However, based on the wide variety of countries the studies come from (three studies from Nigeria, two studies each from India and Iran, one study each from Jamaica, Norway, Saudi-Arabia, Türkiye, and UK), we conclude that HBB is equally safe across ethnicity in the labor/childbirth indication. The following comment on ethnicity was added to the main text (end of section 5.3): “While only one report described data on ethnicity, the fact that the studies came from many countries (three studies from Nigeria, two studies each from India and Iran, one study each from Jamaica, Norway, Saudi-Arabia, Türkiye, and UK), we conclude that HBB is equally safe across ethnicities in the labor/childbirth indication.”

Information on patient history was not provided in any of the references.

Mean or median maternal age was 22-30 years in all studies with two not providing either value and one only mentioning the % of women in certain age brackets. Thus, we considered that adding this information to Table 1 will not be informative. Rather we inserted the following more general statement into the legend of Table 1: “Mean/median age of the mother ranged from 22 to 30 years.”

We found your question on parity interesting: while three articles did not provide corresponding information, the others ranged from being conducted in only nulliparous to those conducted only in multiparous women. Thus the information on parity was added as a new column to Table 1. The following comment on ethnicity was added to the main text (end of section 5.3): “Overall, HBB has been studied in both nulli- and multiparous women (Table 1).”

Comment 7: The literature reports that cases of HBB toxicity are primarily unintentional (BMJ Case Rep. 2020;13(2):e234029; Journal of the Royal Society of Health. 1997;117(4):242-244). However, within the authors' literature search, were there any studies showing fatal outcomes due to standard HBB use? If so, this should be clarified in the manuscript.

Reply 7: Thank you for pointing us to these two articles. We have added the article from BMJ Case Reports to section 6 where we now state “However, a case report describes a 4-year old girl who accidentally ingested 12 tablets of 150 µg each and presented to an emergency department with hyperactivity and confusion; the patient improved and could be discharged after 2 days at the hospital without lasting consequences [88]. Other than such overdose, no cases of severe adverse event were found in our searches.” Unfortunately, we could not identify the article from the “Journal of the Royal Society of Health” because our searches did not find any journal by this name.

Comment 8: Finally, this reviewer would like the authors to expand some of the scope or provide a clarifying summary table that could guide additional direct questions requiring prioritization of future research and identify key knowledge gaps that could be addressed.

Reply 8: You touche on a question that we gave plenty of thought in the writing phase of the manuscript. As can be seen from the manuscript, other than for the off-label indication of labor/childbirth surprisingly little high-quality data is available from the published literature. Therefore, we decided to add a schematic diagram to section 6 instead. Thus, the key knowledge gap is an only small number of RCTs in approved indications; a need for more RCTs is the final conclusion of the originally submitted manuscript.

Reviewer 2 Report

Comments and Suggestions for Authors

Dear Authors,

I would like to thank you for manuscript entitled " A comprehensive review of the effects of hyoscine butylbromide in childhood ".

I have few comments:

1- while writing the review article manuscript try to make it as an interesting story. repeating the word of investigator make the context boring. please improve your writing.

2- can you add a table about the clinical trials performed on HBB?, please check clinicaltrials.gov website.

3- It would be catchy if you make a representative figure for the mechanism of action of HBB.

4- The benefits of HBB is enormous, but suppose to have side effects, which is logic similar to any medicine. Please make a new heading discussing the side effects.

Thank you

Author Response

General comment: I would like to thank you for manuscript entitled " A comprehensive review of the effects of hyoscine butylbromide in childhood ". I have a few comments:

Reply: We thank you for your time and effort in reviewing our manuscript and are happy to learn that you generally liked our manuscript. We reply to the specific comments as follows.

Comment 1: While writing the review article manuscript try to make it as an interesting story. repeating the word of investigator make the context boring. please improve your writing.

Reply 1: Thank you for this comment. We fully agree that a story is more interesting to read than a listing of observations. On the other hand, recent years have seen growing concerns about bias in reporting data. It was our goal to comprehensively describe and, where needed, critique the available data in the field. We can see how for instance a systematic descriptions where studies came from may be boring; on the other hand, reviewer 1 explicitly asked us to add data on ethnicity to Table 1. While specific data on ethnicity are exceedingly rare, systematically mentioning country of origin of a study is a good secondary solution; as the studies come from so many countries, this would support that ethnicity is not a relevant factor in the efficacy and tolerability of HBB.

Comment 2: Can you add a table about the clinical trials performed on HBB?, please check clinicaltrials.gov website.

Reply 2: Given the large number of indications in which HBB has been investigated and the very small number of studies within each indication, we feel that a table listing all studies (more than 60 studies covered in the manuscript) is not helpful. The only exception is the labor/childbirth indication where we actually provided a table of all studies. Based on a suggestion of reviewer 1, this table has been expanded in the revised version.

Checking clinicaltrials.gov for ongoing trials is a very good suggestion. A search of clinicaltrials.gov with the key word combination “hyoscine butyl bromide” or “buscopan” for intervention and a limitation to trials performed/recruiting participants aged 17 or younger (the scope of our manuscript) yielded three hits. One of them started recruitment in 2017 and completed recruitment in 2018; this study has been reported and was already included in our manuscript as reference #17. A trial from Israel evaluated a role for HBB in the treatment of organophosphate poisoning. According to clinicaltrials.gov, this study was expected to complete recruitment in 2009. However, no results were reported and the trial is currently listed as “withdrawn” on clinicaltrials.gov. Finally, a study from France is comparing the role of HBB and anabotulinum toxin A in children with cerebral palsy. According to clinicaltrials.gov, this study is expected to be completed in August 20025. In conclusion, checking clinicaltrials.gov did not reveal any completed studies not already covered in the manuscript and only one ongoing trial. A comment to this effect was added to section 5. It reads “Unfortunately, based on clinicaltrials.gov (assessed on 5.4.2025) only one study of HBB in children is currently ongoing; this trials compares HBB with anabotulinum toxin A for the treatment of drooling in children with cerebral palsy (NCT03616067).”.

Comment 3: It would be catchy if you make a representative figure for the mechanism of action of HBB.

Reply 3: We respectfully disagree with this comment because the mechanism of action is out of scope for our manuscript as focus on uses in infants, children and adolescents. Moreover, the very first sentence of the Introduction already summarized the mechanism of action and referenced this for readers wishing to learn more about it.

Comment 4: The benefits of HBB is enormous, but suppose to have side effects, which is logic similar to any medicine. Please make a new heading discussing the side effects.

Reply 4: Thank you for this suggestion. Information on safety and tolerability of HBB was included in the description of each study reporting such data (some articles, particularly from the early studies unfortunately did not provide safety data. In all cases where data were presented, these were reported in the manuscript. Unfortunately, in most cases these were statements that tolerability was similar as in the control group. Only very few studies provided quantitative data. Based on this very heterogenous and in many cases poor reporting on safety and tolerability, a dedicated chapter on this did not appear useful to us. The only exception again being the studies on the use of HBB in labor/childbirth. The corresponding section on this use is entirely focused on safety as the efficacy target in this indication is the mother, not the baby. A dedicated table on safety of the baby (reflected by the APGAR score) had already been included in the manuscript as Table 1. Moreover, section 6 is largely a discussion of the safety of HBB in pediatric uses.

Round 2

Reviewer 1 Report

Comments and Suggestions for Authors

The authors correctly addressed most of the observations regarding their manuscript submitted to JCM. There are multiple reviews in the literature on the effects of HBB for treating abdominal pain and other symptoms. However, the effects of HBB in children is a topic that has rarely been addressed in a comprehensive and organized manner, as the authors did in this article. Therefore, its publication is justified.

Prior to acceptance, I would like to clarify two points:

First: The second reference provided by this reviewer corresponded to the following study: https://journals.sagepub.com/doi/10.1177/146642409711700409?icid=int.sj-abstract.similar-articles.1 (https://pubmed.ncbi.nlm.nih.gov/9375488/).

Second: I would like to return to the point about access to the information sources used for this review. Among their responses, the authors state the following: "While this response may be frustrating, we will happily make the full texts we have available to the reviewer offline if he/she provides us with a list he/she specifically would be interested in."
Therefore, I suggest that in the section "Data Availability Statement: No new data was created or analyzed in this study. Data sharing is not applicable to this article," it be mentioned that the articles used in this work, regardless of their source, may be provided or shared by the authors upon reasonable request.

Author Response

General comment: The authors correctly addressed most of the observations regarding their manuscript submitted to JCM. There are multiple reviews in the literature on the effects of HBB for treating abdominal pain and other symptoms. However, the effects of HBB in children is a topic that has rarely been addressed in a comprehensive and organized manner, as the authors did in this article. Therefore, its publication is justified.

Reply: We are grateful for your positive evaluation of our manuscript and thank you for the time spent reading it.

Reviewer comment #1: The second reference provided by this reviewer corresponded to the following study: https://journals.sagepub.com/doi/10.1177/146642409711700409?icid=int.sj-abstract.similar-articles.1 (https://pubmed.ncbi.nlm.nih.gov/9375488/).

Reply #1: Thank you for providing the link. We were now able to look at this reference. Importantly, this reference is about hyoscine, not about hyoscine butylbromide. The big difference is that hyoscine has good penetration into the brain, whereas hyoscine butylbromide (a quaternary amine) does not (s. line 37 in manuscript). Therefore, this case report is not applicable to our manuscript.

Reviewer comment #2: I would like to return to the point about access to the information sources used for this review. Among their responses, the authors state the following: "While this response may be frustrating, we will happily make the full texts we have available to the reviewer offline if he/she provides us with a list he/she specifically would be interested in."
Therefore, I suggest that in the section "Data Availability Statement: No new data was created or analyzed in this study. Data sharing is not applicable to this article," it be mentioned that the articles used in this work, regardless of their source, may be provided or shared by the authors upon reasonable request.

Reply #2: We have added the suggested statement but for legal reasons had to add “as far as copyright law allows”.

Reviewer 2 Report

Comments and Suggestions for Authors

Dear Authors,

I would like to thank you for the improvements you did in the manuscript. I believe it is now ready to go.

Thank you.

Author Response

As no additional points were raised by the referee, no additional reply is made.